# Catabolism of acetosyringone and co-metabolic transformation of 2,4,6-trichlorophenol by a novel FAD-dependent monooxygenase

Tomas Engl,[1] Lydie Jakubova,[1] Zdena Skrob,[2] Stephanie Campeggi,[1] Roman Skala,[1] Magdalena Folkmanova,[1] Petr Pajer,[3] Martin Chmel,[3,4] Tomas Cajthaml,[2] Michal Strejcek,[1] Jachym Suman,[1] Ondrej Uhlik[1]

**ABSTRACT**  Acetosyringone (AS), a prototypical syringyl-type monomer of lignin, functions as a model compound for the study of microbial catabolism of S-lignin-derived aromatics. In this study, we present the discovery of a novel metabolic pathway for AS catabolism, initiated by a previously uncharacterized FAD-dependent oxidoreductase, designated AsdA. In contrast to the sole previously documented AS funneling route, which entails side chain modification and conversion to syringic acid, AsdA catalyzes direct hydroxylation of the aromatic core. This represents a mechanistically distinct entry into central metabolism. The identification of this enzyme was achieved through metagenomic and functional analyses of a bacterial consortium enriched on AS as the sole carbon source. The consortium, predominantly comprising *Pseudomonas rhizophila*, exhibited co-metabolic transformation of the chlorinated pollutants 2,4,6-trichlorophenol (2,4,6-TCP) and 2,6-dichlorophenol. Subsequent functional assays substantiated the hypothesis that AsdA facilitates the transformation of both AS and 2,4,6-TCP. Induction assays employing a biosensor strain derived from the bacterial isolate *Pseudomonas rhizophila* AS1 confirmed AS-specific upregulation of the *asd* gene cluster. A survey of publicly available metagenomes has revealed that *asdA* is narrowly distributed but enriched in rhizosphere environments, pointing to its ecological significance. In summary, the present study unveils a hitherto unrecognized route for AS transformation and identifies an enzyme that exhibits dual functions in lignin-derived aromatic catabolism and environmental pollutant transformation. While the mechanisms underlying TCP degradation are well-established, the specific enzyme responsible for the conversion to 2,6-dichloro-*p*-hydroquinone had remained elusive—a knowledge gap that has now been addressed by AsdA.

**IMPORTANCE** The microbial conversion of lignin monomers is central to the global carbon cycle, yet pathways for syringyl-derived aromatics remain poorly resolved. Here, we identify AsdA, an enzyme initiating a previously unrecognized route for acetosyringone catabolism, providing new insight into how this abundant plant-derived compound is integrated into microbial metabolism. Beyond expanding the mechanistic diversity of lignin degradation, AsdA also catalyzes a key step in the transformation of the chlorinated pollutant 2,4,6-trichlorophenol, linking natural and anthropogenic compounds within a shared metabolic framework. The restricted yet rhizosphere-enriched distribution of *asdA* underscores its specialized role in plant–microbe interactions. By integrating enzyme function, microbial community context, and metagenomic distribution, we demonstrate how a single catalytic activity connects metabolic pathways and ecosystem processes, illustrating a multi-scale systems biology perspective on aromatic compound turnover.

**Peer Reviewer** Celso Martins, Universidade Nova de Lisboa, Oeiras, Portugal

Address correspondence to Ondrej Uhlik, ondrej.uhlik@vscht.cz.

The authors declare no conflict of interest.

See the funding table on p. 17.

**KEYWORDS** secondary plant metabolites (SPMs), acetosyringone (AS), FAD-dependent monooxygenase AsdA, chlorophenols (CPs), 2,4,6-trichlorophenol (2,4,6-TCP), degradation, co-metabolism, metagenome-assembled genomes (MAGs), aromatic pollutants (APs)

Lignin, a high-molecular-weight aromatic heteropolymer, is an abundant structural component in vascular plants. During microbial degradation, lignin releases numerous phenolic compounds (1). Its macromolecular structure consists of sinapyl- and syringylphenylpropanoid analogs (S-lignin), coniferyl- and guaiacylphenylpropanoids (G-lignin), and *p*-coumarylphenylpropanoids (H-lignin) (2, 3). Despite its structural rigidity and resistance to degradation, many microorganisms can contribute to the degradation of lignin (1). While fungi were historically considered the primary lignin degraders, recent studies have emphasized the role of bacteria, which utilize diverse enzymatic machinery to depolymerize lignin and metabolize its aromatic units as carbon and energy sources (4, 5). However, the topic of bacterial lignin degradation remains poorly explored, leaving many scientific questions unanswered and generating further hypotheses. Advances in multi-omics approaches, such as (meta)genomics and (meta)transcriptomics, along with metabolomic analyses, may help unravel many of the unknowns associated with bacterial lignin degradation (1, 4, 5).

Acetosyringone (AS), a phenolic compound, is derived from S-lignin, a structural polymer that is present in large quantities in hardwoods and other vascular plants. AS is a typical syringyl-type monomer and a model compound for studying the microbial catabolism of S-lignin-derived aromatics. Additionally, AS acts as a signaling molecule that plays a crucial role in plant-bacteria communication (2, 6, 7). Despite its ecological relevance and structural similarity to various aromatic pollutants (APs), the microbial degradation pathways of AS remain only partially understood (8). Among the few extensively studied S-lignin degraders is *Sphingobium lignivorans* SYK-6, whose genome encodes enzymes involved in sinapic acid catabolism, such as feruloyl-CoA hydratase/synthetase (FerA/FerB), syringaldehyde dehydrogenase (DesV), and tetrahydrofolate-dependent *O*-demethylase (DesA) (8–12). Recent research identified the AvcABCDEF catabolic pathway in *Sphingobium lignivorans* SYK-6, which transforms acetovanillone and AS into vanillate and syringate, respectively (13). However, the catabolism of AS and other S-lignin substructures is likely to involve additional, as yet undescribed, enzymes and pathways (8).

Importantly, AS is structurally similar to chlorophenols (CPs), which are synthetic compounds widely used in industrial and agricultural applications that have been classified as priority pollutants by the US Environmental Protection Agency (14) due to their persistence, toxicity, and ability to bioaccumulate (14–17). The structural similarity between AS and other naturally occurring phenolic compounds and CPs implies that enzymes capable of degrading lignin or its derivatives may also facilitate the biodegradation of CPs, in agreement with the co-metabolism hypothesis (18–22). To explore this connection, we enriched a soil bacterial consortium capable of utilizing AS as the sole carbon source. This resulting consortium, dominated by *Pseudomonas rhizophila*, efficiently depletes CPs, more specifically 2,4,6-trichlorophenol (2,4,6-TCP) and 2,6-dichlorophenol (2,6-DCP). Its simple taxonomic composition makes it an ideal model for studying the genes and pathways involved in lignin degradation and for identifying degradation intermediates. In this study, we reveal a hitherto unknown gene cluster in *Pseudomonas rhizophila* AS1 that initiates AS downstream funneling and promotes the co-metabolic transformation of 2,4,6-TCP.

## RESULTS

### Depletion of APs, including CPs, by a consortium enriched on AS

An enrichment culture was initiated by inoculating compost soil (23) into a mineral medium (MM) supplemented with AS as the sole carbon source, with 5% (vol/vol) inoculum being repeatedly transferred into fresh medium every 1–2 weeks over 6

months. Passaging was performed once visible growth occurred. Over successive transfers, 16S rRNA gene sequencing (data not shown) revealed a progressive reduction in community complexity while still utilizing AS as the sole carbon source. After 23 passages, the enrichment was designated ASC12 and used for subsequent analyses.

The ASC12 consortium's potential to deplete selected APs and CPs was assessed using resting cell assays. The results showed that, although dibenzofuran, diphenyl ether, naphthalene, and 2-CP were depleted at statistically significant rates, their relative depletion rates remained below 10% over 48 h. By contrast, 2,4,6-TCP and 2,6-DCP were depleted by 48 and 35%, respectively. Based on previous screening results, where compounds were tested in pooled mixtures, phenol and CPs were selected for follow-up single-compound experiments. In these assays, phenol was completely depleted, and, more notably, both 2,4,6-TCP and phenol were completely depleted, while 2,6-DCP was partially depleted at 46.6% (Fig. 1).

## Taxonomic and functional characterization of the ASC12 consortium

An analysis of 16S rRNA amplicons from the ASC12 consortium was performed to assess the taxonomic stability of the consortium during passaging. This analysis revealed low taxonomic complexity, with amplicon sequence variants affiliated with *Pseudomonas* and *Methylotenera* dominating. Given the low complexity of the consortium, it was reasonable to sequence its entire metagenome. From the obtained metagenomic data, we assembled three circular MAGs, one of which corresponds to a plasmid (accession numbers CP182238.1, CP182239.1, CP182240.1) (Fig. S1). The genomes were classified in the GTDB system as s__*Pseudomonas_E rhizophila* and s__*Methylotenera_A versatilis_A*, which correspond to *Pseudomonas rhizophila* and *Methylotenera versatilis* in the NCBI taxonomy. Based on genome coverage, the ratio of the *Pseudomonas* and *Methylotenera* members was 2:1.

The constructed database of coding sequences of genes involved in bacterial catabolism of S-lignin or degradation of CPs, referred to as ASD_DB, contains sequences ranging from proteins initiating AS catabolism to *O*-demethylases, as well as proteins involved in degradation of 2,4,6-TCP, 2,6-DCP, and other CPs. In addition, ASD_DB contains oxygenases, which are key enzymes in the initial steps of aromatic contaminant transformation. Using this database, homologous sequences within the ASC12 consortium metagenome were identified by BLASTP. These sequences were predicted to play a potential role in the transformation of AS as well as the co-metabolism of 2,4,6-TCP and 2,6-DCP. As shown in Fig. 2, the most promising gene hits were assigned to the *Pseudomonas* MAG, while no significant hits were found in the *Methylotenera* MAG. This

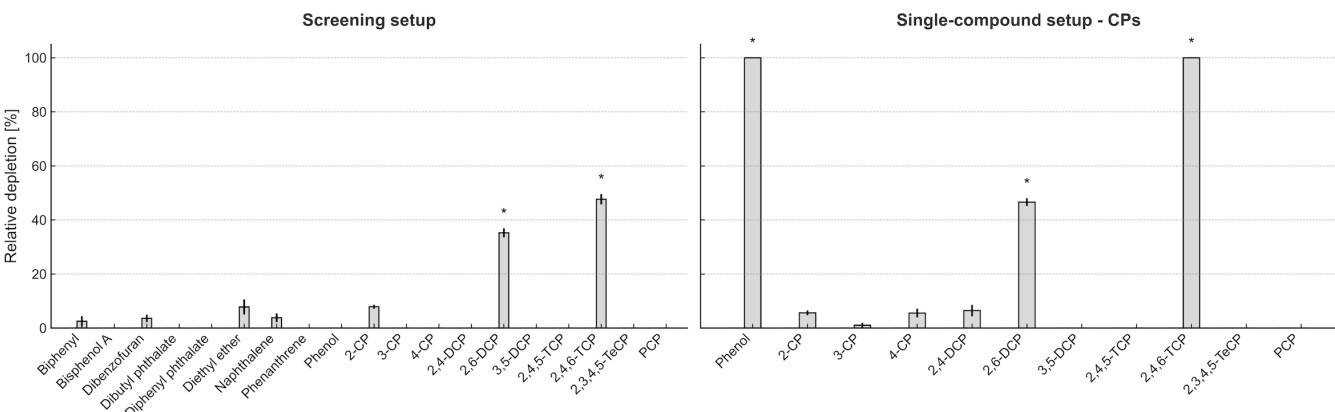

**FIG 1** Depletion of selected APs and CPs by the bacterial consortium ASC12. Results in which the arbitrary significant depletion threshold (>10%) was exceeded are marked with an asterisk. The percentage of substrate depletion by the live ASC12 consortium, compared to the percentage by the autoclaved cells, is shown. On the left are results from the screening setup, in which selected pollutants were pooled into mixtures to reduce the number of samples; on the right are results from the single-compound setup, where each CP was tested separately.

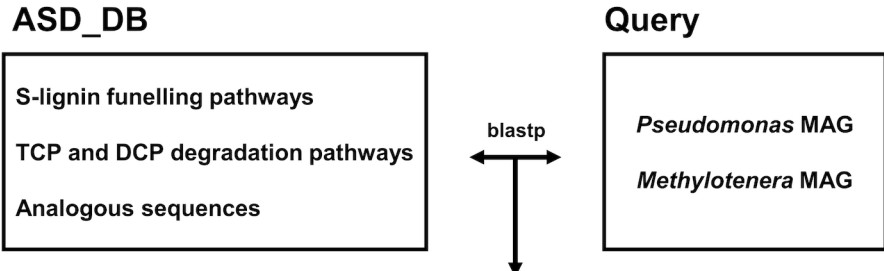

FIG 2 Schematic workflow for identifying coding sequences involved in AS catabolism and CP transformation in ASC12 MAGs. The ASD_DB (a custom database of genes involved in bacterial catabolism of S-lignin-derived compounds or degradation of CPs) was first compared with annotated ASC12 MAGs using BLASTP. Based on functional annotation and amino acid (AA) identity, the most promising hits were selected. Locus tags starting with "ACNJA3" indicate that the sequence originates from a *Pseudomonas* MAG.

suggests that *Pseudomonas* within ASC12 plays the major role in AS depletion as well as the co-metabolism of CPs.

## *Pseudomonas rhizophila* AS1 is responsible for the depletion of 2,6-DCP and 2,4,6-TCP

Cultivation of the ASC12 consortium members on solid MM supplemented with 100 mM AS as the sole carbon source enabled the isolation of a pure culture designated AS1. The affiliation of the strain AS1 to *Pseudomonas rhizophila* was confirmed by matrix-assisted laser desorption ionization-time of flight mass spectrometry (MALDI-TOF MS) and 16S rRNA gene sequencing.

The isolated *P. rhizophila* AS1 strain exhibited the ability to utilize AS as a sole carbon source. There was no significant difference in the growth curves of the AS1 strain and the ASC12 consortium (data not shown). The role of the AS1 strain in the depletion of both 2,6-DCP and 2,4,6-TCP was confirmed by resting-cell assays (RCAs) when grown on AS (Fig. 3). Notably, the depletion of CPs occurred only when AS1 was cultured in the presence of AS; no depletion was detected when cultured on LB medium, supporting the dependence of CP depletion on the metabolism of AS.

## Identification of the gene responsible for the transformation of AS and 2,4,6-TCP

The sequences of the most promising hits identified using the constructed ASD_DB were successfully cloned into *Escherichia coli* DH5α cells, and the activity of the expressed genes was confirmed by the ability of the respective clones to deplete the selected control substrates, which were chosen based on prior database annotations. However, among the clones tested, only one was able to deplete both AS and

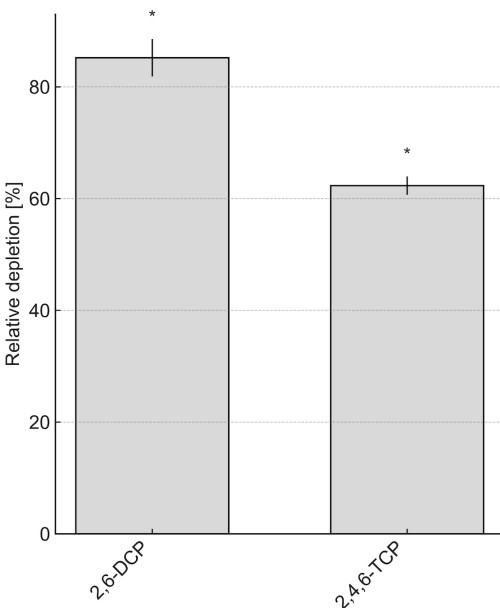

**FIG 3** Depletion of 2,6-DCP and 2,4,6-TCP by *Pseudomonas rhizophila* AS1 precultivated in MM with AS as the sole carbon source. The percentage of substrate depletion by the live AS1 cells, compared to the percentage by the autoclaved cells, is shown. Results in which the arbitrary significant depletion threshold (>10%) was exceeded are marked with an asterisk.

2,4,6-TCP (Table 1), specifically the clone carrying the gene *ACNJA3_12680*, the product of which showed 41% amino acid similarity to the *p*-nitrophenol 4-monooxygenase (PnpA) from *Pseudomonas* sp. WBC-3. The identified *ACNJA3_12680* gene was annotated as an FAD-dependent oxidoreductase and designated *asdA* (AS degradation). Thus, our results suggest that this FAD-dependent oxidoreductase, AsdA, initiates the AS downstream funneling and is responsible for the co-metabolic transformation of 2,4,6-TCP.

The amino acid sequence of the AsdA was queried against the nr-protein database using BLASTP, which highlighted its distinctness. The highest similarities were observed with an FAD-dependent oxidoreductase from *Pseudomonas mediterranea* S58 (WP_159285523.1) and FAD-dependent monooxygenases from *Pseudomonas* sp. Pf153 (WP_082339948.1) and *Pseudomonas* sp. B21-056 (WP_265067184.1) with 97.84%, 96.88%, and 71.15% amino acid similarity, respectively. In addition, the published research associated with these sequences has not reported any involvement in the transformation of lignin monomers or related compounds (CPs), and, in general, their functions have not been previously described in the literature. All other sequences

**TABLE 1** Screening of the enzymatic activity of candidate gene products toward 2,6-DCP and 2,4,6-TCP[a]

| Cloned gene | Depletion (control substrate) | 2,6-DCP | 2,4,6-TCP |
|---|---|---|---|
| ACNJA3_13350 | + (vanillin) | − | − |
| ACNJA3_13355 | + (ferullic acid) | − | − |
| ACNJA3_13345 | + (Ferullic acid) | − | − |
| ACNJA3_13370 | + (vanillate) | − | − |
| ACNJA3_13365 | + (vanillate) | − | − |
| ACNJA3_06770 | + (protocatechuate) | − | − |
| ACNJA3_06775 | + (protocatechuate) | − | − |
| ACNJA3_12025 | + (dihydroxyphenylacetate) | − | − |
| **ACNJA3_12680** | **+ (acetosyringone)** | − | + |

[a]The activity of each expressed enzyme was verified by its ability to transform a control substrate selected based on its functional annotation. The gene responsible for the transformation of both AS and 2,4,6-TCP is highlighted with boldface. Locus tags beginning with "ACNJA3" suggest that the sequence originates from a *Pseudomonas* MAG.

showed less than 56% amino acid similarity. The phylogenetic tree constructed from the 100 sequences most similar to AsdA (Fig. 4) also shows that FAD-dependent oxidoreductases form distinct clusters of sequences with high internal similarity. However, AsdA does not belong to any of these clusters and may represent a new phylogenetic branch of FAD-dependent oxidoreductases.

Comparative analysis of the genomic regions surrounding the *asdA* gene and its closest homologs, conducted using Clinker, revealed homologous gene architectures exclusively in clusters from *Pseudomonas* sp. Pf153, *P. mediterranea*, and *Pseudomonas* sp. B21-056, while other gene clusters displayed no similarity in their genomic organization (Fig. 5). The genes surrounding *asdA* within the AS1 genome were designated as the *asd* gene cluster and further characterized as follows: alpha/beta hydrolase I, ACNJA3_12690 (*asdB*); VOC family protein, ACNJA3_12685 (*asdC*); alpha/beta hydrolase II, ACNJA3_12675 (*asdD*); and hypothetical protein (transporter), ACNJA3_12670 (*asdE*).

## The *asdA* gene is induced by AS

For clones carrying individual gene sequences from the *asd* gene cluster, RCA and subsequent analysis of CP depletion by HPLC revealed that AS and 2,4,6-TCP were depleted only by the clone carrying *asdA* and not by any other clone carrying other genes from the *asd* gene cluster (Fig. S3). This implies that only AsdA is involved in the initial transformation steps of AS and 2,4,6-TCP. Considering this and the fact that *P. rhizophila* AS1 did not deplete 2,4,6-TCP or 2,6-DCP except when grown on AS, AS was suspected to be an inducer of the *asd* gene cluster, allowing co-metabolic depletion of CPs. To study the induction of *asd* genes *in vivo*, an ASC12-derived biosensor strain was prepared by introducing the pTr-IR-egfp plasmid into wild-type AS1 cells. Of all the substrates tested, AS and orcinol induced eGFP fluorescence (Fig. S2). Specifically, the induction rates in the biosensor strain were 2- and 3-fold higher at 2 mM and 3 mM AS, respectively, compared to the wild-type strain. Meanwhile, the same concentrations of orcinol (2 and 3 mM) determined induction rates in the biosensor strain that were 1.2- and 1.4-fold higher than in the wild-type strain (Fig. 6).

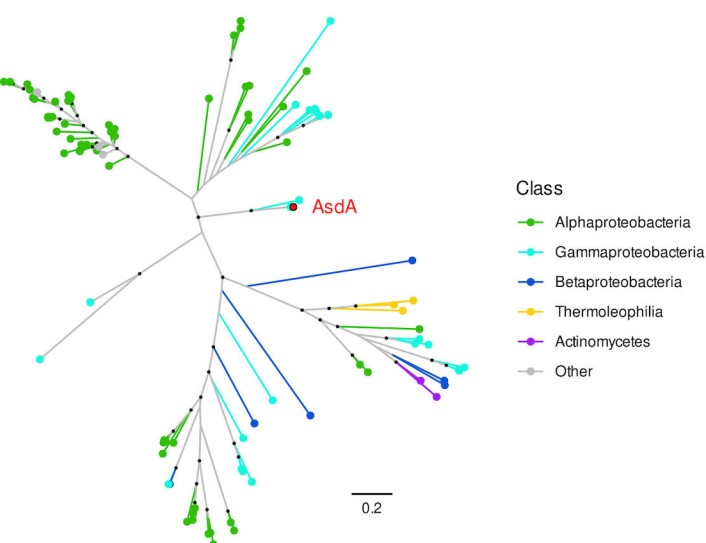

**FIG 4** Phylogenetic tree of the 100 most similar AsdA homologs based on amino acid sequence alignment. Clusters of FAD-dependent oxidoreductases are indicated, with AsdA (highlighted in red) to show its distinct position outside of established clades. Black dots indicate significant branch support by ultrafast bootstrap approximation UFBoot > 95. Branch length indicates the number of substitutions per site.

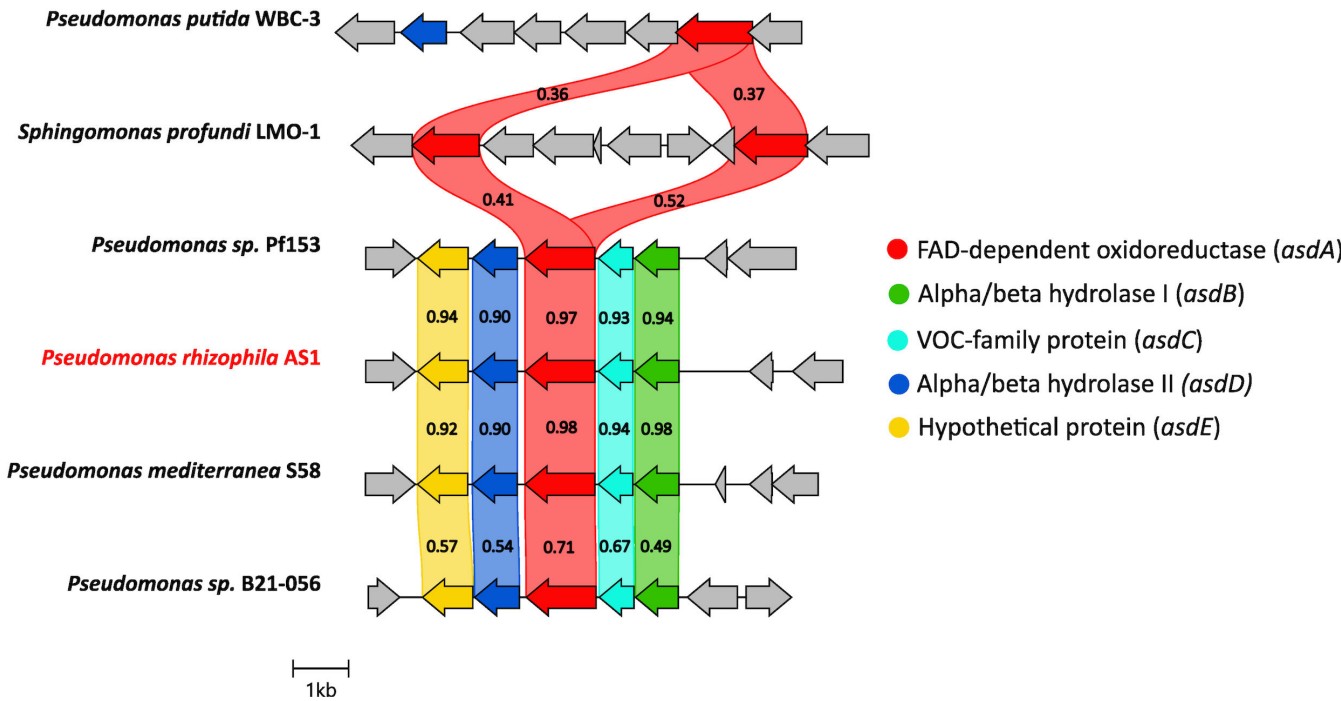

**FIG 5** Architecture of the *asd* gene cluster in *P. rhizophila* AS1 (highlighted in red) and comparison with similar clusters containing a gene homologous to *asdA*.

## LC-MS analysis of AS and 2,4,6-TCP transformation products

For the LC-MS analysis of the AS and 2,4,6-TCP transformation products, two strains of *E. coli* BL21(DE3) were used: one carrying only the *asdA* gene and the other carrying the entire *asdBCAD* gene cluster. The analysis revealed a molecular feature with $m/z$ = 213.0771, which was found exclusively in the samples with *asdA*-bearing cells

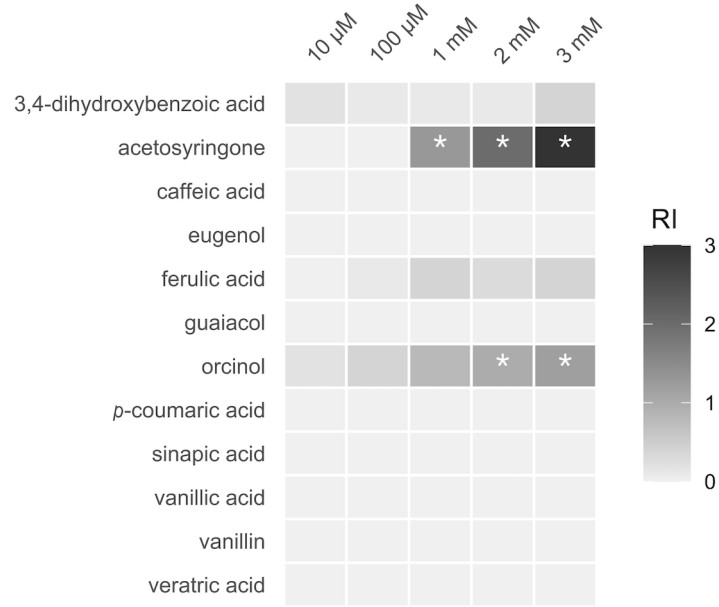

**FIG 6** Heat map of the induction assays employing the AS1/eGFP and AS1/wild-type strains. The relative induction rate (RI) represents the difference between the induction rates of the two strains, with RI = 0 denoting no difference. An asterisk indicates an RI value at which the AS1/eGFP strain had an induction rate that was at least onefold higher than the AS1/wild-type strain.

(Fig. 7). This compound was not detected in samples from cells harboring the entire *asdBCAD* gene cluster (Fig. S4). Based on the *m/z* value and the isotopic pattern, the molecular formula of the corresponding compound was determined to be $C_{10}H_{12}O_5$. Subsequent MS/MS fragmentation analyses suggested that the structure corresponds to 2,4-dihydroxy-3,5-dimethoxyacetophenone (AS-OH). The hydroxylation position in AS-OH was inferred from a high-resolution MS/MS analysis. The observed fragment ions and their neutral losses were consistent with side-chain cleavage while retaining the 3,5-dimethoxy-4-hydroxy substitution pattern, indicating that the additional hydroxyl group is in the ortho position on the aromatic ring. Alternative hydroxylation positions were incompatible with the fragmentation pattern (Fig. 7).

Analysis of 2,4,6-TCP transformation products identified two molecular features unique to both *asdA*- and *asdBCAD*-bearing cells (Fig. S5). The major feature had *m/z* = 176.9558 in [M-H]⁻ and was also detected in a dimeric form [2M-H]⁻ with a mass of 356.9079. Its isotopic pattern suggested a molecular formula of $C_6H_4Cl_2O_2$. Based on MS/MS fragmentation analysis, the compound was identified as 2,6-dichlorohydroxyquinone (DCHQ). This assignment was further supported by a spectral match in the Metlin library, achieving a score of 95.78 out of 100. The second feature, with *m/z* = 191.9746 [M-H]⁻, had an isotopic pattern suggesting a molecular formula of $C_7H_6Cl_2O_2$. Based on subsequent MS/MS fragmentation analysis, the compound was proposed to be a 2,6-dichloro-4-methoxyphenol (DCMP). The structural assignment was further supported by a spectral match in the Metlin library, achieving a score of 99.72 out of 100. These results led us to conclude that AsdA acts as an FAD-dependent monooxygenase, initiating the transformation of 2,4,6-TCP, leading to the formation of DCHQ. A minor fraction of DCHQ is likely to be further *O*-methylated to DCMP by the expression host *E. coli* BL21(DE3) (Fig. 8).

## Ecological relevance of the *asdA* gene

A set of 1,007 species-representative Pseudomonadaceae genomes that were available in GTDB was screened for AsdA homologs using the AsdA amino acid sequence as a query. This screening yielded a single hit that exhibited 71.2% identity. A more extensive screening of all 7,438 genomes from the genus *Pseudomonas_E* yielded two hits with >90% identity to *asdA*, one in each of *Pseudomonas mediterranea* and *Pseudomonas uvaldensis*. However, none of the 18 *Pseudomonas rhizophila* reference genomes contained the gene, and the limited number of positive hits among eleven *P. mediterranea* and three *P. uvaldensis* genomes suggests that *asdA* is not widely distributed within these species.

Using the Sandpiper database, 329 metagenomes were identified as containing *Pseudomonas rhizophila*. These metagenomes were subsequently screened for the presence of *asdA*, with a specific focus on the region between bases 205 and 285 within the *asdA* gene, to circumvent the potential cross-detection of homologs. The *asdA* was detected in 9 of the 329 metagenomes, primarily from rhizosphere environments. These included metagenomes from the rhizospheres of potato, tomato, and tobacco. Notably, *asdA* was detected even in low-abundance *P. rhizophila* populations, provided sufficient coverage was attained, suggesting that the gene may be present in low-abundance populations. To further assess the distribution of *asdA*, an additional 3,508 rhizosphere- and compost-derived metagenomes were screened (up to 10 Gb each). Two additional *asdA*-harboring metagenomes were identified: one from the rhizosphere of *Capsicum annuum* treated with *Streptomyces pactum* Act12 (SRR21515679), and one from paddy soil (SRR25078404). Within the examined metagenomes, *asdA*-positive reads were primarily detected in rhizosphere and compost samples, that is, in plant-associated or organic-matter-rich environments. However, the small number of hits and biased sampling prevent a rigorous statistical assessment of habitat enrichment. Considering this ecological association and AsdA's role in AS transformation, we further characterized the metabolic potential of strain AS1, which harbors *asdA*, for its ability to deplete a panel of aromatic compounds associated with S-lignin. Strain AS1 depleted several

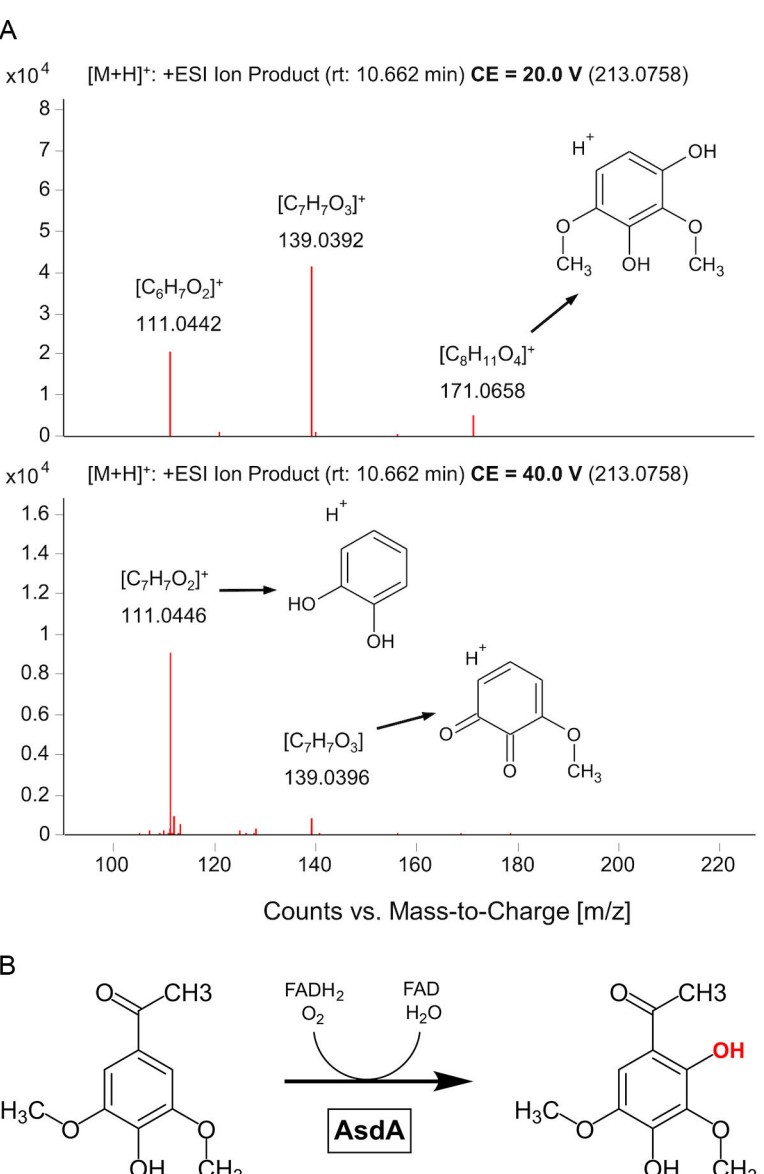

**FIG 7** (A) MS/MS fragmentation spectra of the [M+H]+ ion of AS-OH (*m/z* 213.0758) acquired at two collision energies: 20 V (top panel) and 40 V (bottom panel). The spectrum at lower collision energy (CE = 20 V) shows product ions at *m/z* 111.0442 (corresponding to the elemental composition [$C_6H_7O_2$]+), 139.0392 ([$C_7H_7O_3$]+), and 171.0658 ([$C_8H_{11}O_4$]+), which result from cleavages of the side chain and the loss of functional groups while preserving the 3,5-dimethoxy-4-hydroxy substitution pattern, which requires that the additional hydroxyl group is introduced at the ortho position (2,6-position). Alternative structures involving side-chain hydroxylation or hydroxylation at other ring positions did not match the observed fragmentation pattern. At higher collision energy (CE = 40 V), more extensive fragmentation is observed, with the same major product ions present but exhibiting different relative intensities. Proposed structures of the main fragments are depicted to illustrate the likely fragmentation pathways. (B) Proposed reaction mechanism in which AsdA acts as an FAD-dependent monooxygenase, initiating AS degradation by hydroxylating the aromatic ring at the ortho position.

compounds, including the previously mentioned AS, as well as several structurally related compounds: caffeic acid, *p*-coumaric acid, ferulic acid, 2,4-dihydroxybenzoic acid, sinapic acid, vanillic acid, and veratric acid. However, strain AS1 did not deplete eugenol, guaiacol, or orcinol. Functional analysis confirmed that AsdA contributes specifically to

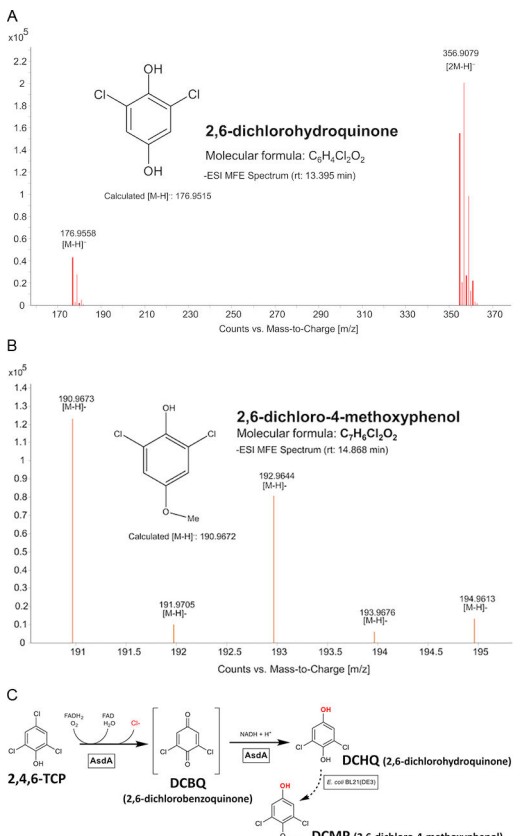

**FIG 8** (A) Determination of the molecular formula and structure of metabolites 2,6-dichlorohydroqui-none (DCHQ) and (B) 2,6-dichloro-4-methoxyphenol (DCMP), produced during 2,4,6-TCP transformation by both *asdA*- and *asdBCAD*-expressing cells, based on MS/MS fragmentation analysis and Metlin database search. (C) Proposed reaction mechanism in which AsdA functions as an FAD-dependent monooxygenase, initiating transformation of 2,4,6-TCP and leading to the formation of DCHQ, which is subsequently *O*-methylated in minor amounts by the expression system *E. coli* BL21(DE3) to DCMP.

the transformation of AS and vanillin, but not to the metabolism of the other tested compounds (Fig. 9).

## DISCUSSION

In this study, we present a newly identified enzyme that initiates the transformation of AS. This enzyme is encoded within a gene cluster that is likely responsible for funneling AS-derived intermediates into downstream metabolism, with broader implications for lignin turnover and plant organic matter (OM) decomposition. Until now, only one metabolic pathway has been known for AS catabolism, in which AS is biotransformed to 3-(4-hydroxy-3,5-dimethoxyphenyl)-3-oxopropanoic acid by the AcvABCDEF enzymes, as observed in *Sphingobium lignivorans* SYK-6 (13). This pathway primarily involves transformations of the AS side chain. In contrast, the FAD-dependent monooxygenase AsdA, which is described in this study, participates directly in the biotransformation of the aromatic core of AS by hydroxylating it. This is consistent with its ability to initiate the co-metabolic degradation of APs, since hydroxylation of the aromatic ring is typically the first step in such degradation (24, 25). CPs were hypothesized to be some such molecules, since 2,4,6-TCP shares structural similarity with AS.

For this study, a liquid culture enrichment strategy was chosen as it is an effective method for isolating degraders of the tested substrates (26). Additionally, compost soil was selected as the initial source for culture enrichment, as it is continuously enriched with OM of plant origin. For these reasons, it was hypothesized that taxa capable of

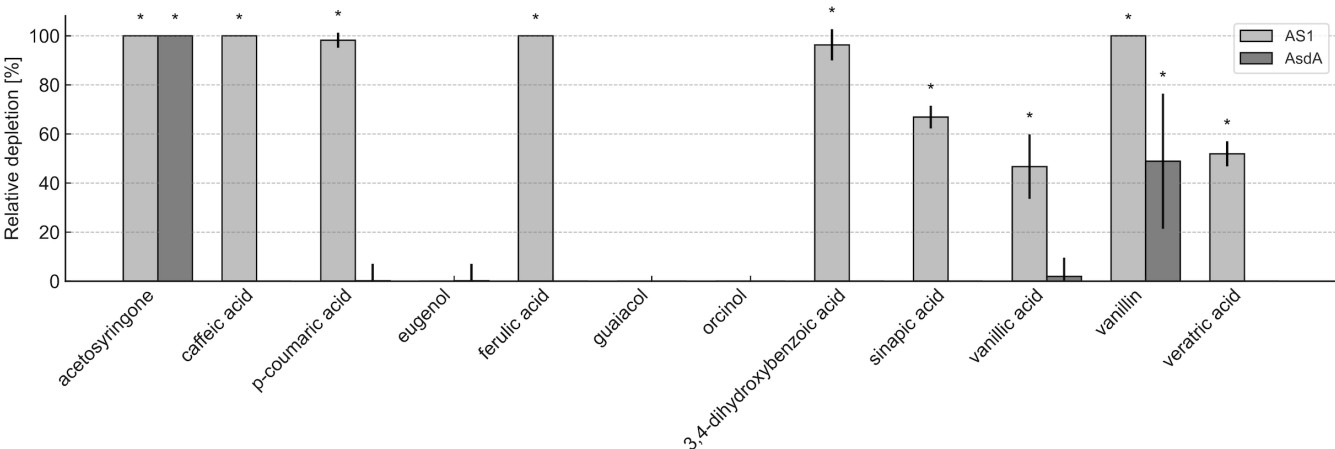

**FIG 9** Depletion of selected S-lignin building blocks by the *P. rhizophila* AS1 and *E. coli* cells expressing the *asdA* gene. Results in which the arbitrary significant depletion threshold (>10%) was exceeded are marked with an asterisk. Relative depletion denotes the percentage decrease in substrate peak area relative to the corresponding control, normalized to the initial substrate concentration.

degrading the structural components of lignin would be present, particularly because OM decomposition in this compost soil occurs at high rates, within a timeframe of several weeks to months. These predictions were confirmed, leading to the isolation of a compositionally streamlined bacterial consortium. Metagenomic analysis, which is significantly more effective in taxonomically less complex matrices (27), along with subsequent cultivation, identified *Pseudomonas rhizophila* AS1 as the dominant taxon responsible for depletion of AS, 2,4,6-TCP, and 2,6-DCP. In doing so, it uses its hitherto undescribed FAD-dependent monooxygenase, AsdA (Table 1; Fig. 4). To further emphasize the leading role of the AS1 strain in the S-lignin funneling pathway, we observed a significant and highly specific induction of the *asd* gene cluster in the presence of AS (Fig. 6). Interestingly, orcinol also induced expression of the *asd* gene cluster, yet AS1 neither grew on nor depleted orcinol in RCAs. This potential discrepancy between transcriptional induction and catabolic activity suggests the regulatory promiscuity of the *asd* promoter region, which appears to respond to structural analogs of AS, even when they are not AsdA substrates. This broad recognition of phenolic signals may enable AS1 to sense the presence of S-lignin-derived aromatics in complex, plant-associated environments while retaining the narrow catalytic specificity of AsdA. A similar phenomenon has been described previously, for instance, by Zubrová et al., who found that, in *Ectopseudomonas alcaliphila* JAB1, a wide range of phenolics and terpenes induced transcription of the biphenyl dioxygenase (BPDO) gene, though not all were depleted by BPDO (28).

Along with *Pseudomonas rhizophila* AS1, *Methylotenera versatilis* MT1 was also detected as part of the ASC12 consortium. As a bacterium specializing in the utilization of C1 compounds, we initially hypothesized that *Methylotenera* might contribute to AS demethylation. However, our findings did not support this hypothesis. Genomic analysis of *Pseudomonas rhizophila* AS1 revealed the presence of genes, for example, vanillate *O*-demethylase (*vanAB*), that enable it to cleave methoxy groups from S-lignin building blocks, such as AS (29). Consistently, we did not detect a substantial increase in AS depletion in the ASC12 consortium compared to the AS1 monoculture under the tested conditions (data not presented), suggesting that AS1 alone is sufficient to support AS-based growth. Nevertheless, we propose that the primary ecological function of *M. versatilis* MT1 within the ASC12 consortium is its ability to detoxify the harmful by-products of AS breakdown. The demethylation of AS and the degradation of lignin and its building blocks may produce toxic C1 by-products such as formaldehyde and methanol, which C1 bacteria exemplified by *Methylotenera versatilis* MT1 are likely to

eliminate. Further investigation of this synergistic interaction is necessary, which requires targeted experiments for confirmation.

Through non-targeted LC-MS analysis of AS transformation products, we observed that AS undergoes hydroxylation to AS-OH in *E. coli* carrying only the *asdA* gene, with no other analytes detected (Fig. 7). However, this was not the case for clones harboring the entire *asd* gene cluster, where AS depletion was observed but no additional products were detected compared to control clones without any gene insertion. These findings suggest that the *asd* gene cluster represents a candidate operon likely involved in the further transformation of AS-OH. However, we did not detect any TCA cycle intermediates attributable to AS, and the complete pathway therefore remains beyond the scope of this study. Further tests (Fig. S3) also indicate that *asdA* is indeed the initiating gene in AS transformation, as none of the other genes in the *asd* gene cluster were able to independently transform AS in any way. However, the complete metabolic pathway and the specificity of individual enzymes remain to be elucidated.

The primary metabolite detected during the transformation of 2,4,6-TCP by clones carrying *asdA* was 2,6-dichlorohydroquinone (Fig. 8). The detection of this metabolite is consistent with previous studies, where this reaction is considered to be the initial step in the degradation of 2,4,6-TCP (30–32). This reaction mechanism has been described for genes such as *tcpA* from *Cupriavidus necator* JMP134 (33, 34), *tdfD* from *Burkholderia cepacia* AC1100 (31), or *hadA* from *Ralstonia pickettii* DTP0602 (35, 36). However, in many metagenomic studies, these initiating genes are not detected despite evidence of 2,4,6-TCP degradation under the given conditions (30, 32, 35). This discrepancy may be due to the limited database of known initial degradation genes, which we expand with the identification of the *asdA* gene. Since 2,6-dichlorohydroquinone was detected in samples containing clones harboring the entire *asdBCAD* gene cluster, we propose that only AsdA is involved in 2,4,6-TCP transformation. We interpret the detection of the minor second product, 2,6-dichloro-4-methoxyphenol (DCMP), as evidence of a cellular detoxification response in the *E. coli* expression system. Since DCMP was detected only in samples where DCHQ was formed, in both *asdA*- and *asdBCAD*-expressing cells, it likely originates from *O*-methylation of the reactive intermediate. This may occur either through a promiscuous activity of endogenous SAM-dependent methyltransferases or via non-enzymatic methylation by intracellular S-adenosylmethionine (SAM), both of which have been observed in *E. coli* and other microbial systems (37, 38).

We investigated homologs of both the *asdA* gene and the entire *asd* gene cluster. We identified only three hits in the entire NCBI non-redundant protein database that exhibited high sequence similarity. These hits were unannotated, and their function has not yet been characterized. The next most similar sequences shared less than 56% similarity. To investigate the ecological significance of the *asdA* gene and the AS1 strain, we assessed the strain's ability to deplete multiple S-lignin structural analogs. The AS1 strain substantially depleted the tested compounds, whereas *asdA* expression alone mainly depleted AS and, to a lesser extent, vanillin (Fig. 9). This indicates that AsdA exhibits high substrate specificity. Furthermore, the *Pseudomonas rhizophila* genome was detected in several hundred rhizosphere and compost metagenomes, but the *asdA* gene itself was only detected in a small fraction of these. This suggests that *asdA* is a rare gene that has likely been overlooked in previous metagenomic studies due to low sequencing depth. Therefore, combining liquid enrichment culturing with targeted metagenomic analysis remains a valuable strategy for uncovering novel pathways such as lignin degradation.

In conclusion, our study describes the discovery of a novel gene cluster involved in the catabolism of AS, which may be integrated into the central metabolism of S-lignin. We describe a phylogenetically distinct FAD-dependent monooxygenase designated AsdA, which initiates AS transformation by directly hydroxylating its aromatic core. This contrasts with the only previously described AS funneling pathway, where side chain modifications convert AS into 3-(4-hydroxy-3,5-dimethoxyphenyl)-3-oxopropanoic acid, subsequently transformed into syringic acid (13). Notably, AsdA also demonstrates dual

functionality, being involved in the transformation of 2,4,6-TCP. While the downstream degradation pathway of 2,4,6-TCP is known in several taxa, metagenomic studies have so far lacked the key gene responsible for its conversion into 2,6-dichloro-*p*-hydroquinone; this critical gap is now filled by the discovery of *asdA* (32, 35). This finding enhances our understanding of the complex relationship between phenolic compounds and environmental pollutants, a topic that has been the subject of numerous studies (25, 28, 39–44). It also highlights the potential of microbial enzymes evolved for lignin degradation to act on CPs and other anthropogenic contaminants by exploiting their broad substrate promiscuity and/or functional exaptation. Identifying AsdA provides further insight into microbial lignin catabolism and demonstrates the evolutionary adaptability of enzymes that transform APs.

## MATERIALS AND METHODS

### Chemicals, solutions, and primers

The composition of the MM and stock solutions used is described in the Supplemenal Material. The sequences of the primers used are provided in Table S1.

### Enrichment of the bacterial consortium ASC12 on AS

The bacterial consortium ASC12 was obtained from garden compost soil, which was produced through the continuous accumulation of organic materials, including grass clippings and fruit and vegetable waste, and collected in the municipality of Mirošovice, Central Bohemia (23). The soil characteristics are provided in Table S2. Before use, the soil was sieved through a 2 mm mesh (23), and 5 g of soil was transferred to a solution of 0.1% $Na_2P_2O_7$ (Sigma-Aldrich, Germany) and agitated on an orbital shaker overnight at 100 RPM and 12°C.

AS-enriched medium was prepared by adding ethanolic AS stock to a flask for a final concentration of 1 mM, air-drying to evaporate ethanol, and adding sterile sea sand as a growth support. Five milliliters of liquid MM in a 50 mL flask was inoculated with 500 µL of an overnight-agitated soil suspension. Cultures were incubated in the dark at 12°C, shaken at 100 RPM, and transferred to fresh AS medium (5% inoculum) every 1–2 weeks. After 6 months, cultures were stored at −80°C as glycerol stock (1:1 with 50% autoclave-sterilized glycerol). Additionally, 100 µL of culture was centrifuged (20,000 RCF, 4°C, 2.5 min) and the pellet was flash-frozen and stored at −80°C for DNA extraction.

### Screening of the transformation potential of the ASC12 consortium

The ASC12 consortium was cultivated in 20 mL MM with 100 µM AS in a 100 mL flask at 12°C, 100 rpm for 48 h, then scaled up to 100 mL MM in a 500 mL flask under the same conditions until the exponential phase. Cells were harvested for RCAs by centrifugation ($6,000 \times g$, 10 min), resuspended in 5 mL sterile MM, pooled, adjusted to 50 mL, and centrifuged again. After two washes, pellets were resuspended in MM to $OD_{600} = 1$. The suspension was split, with one portion kept on ice and the other autoclaved as a negative control.

For RCAs, 250 µL of the cell suspension (or negative control) was transferred to 4 mL sterile glass tubes. To each tube, 2.5 µL of a 10 mM solution of APs and/or CPs was added. Tubes were incubated for 48 h at 12°C with shaking at 100 rpm. All samples were prepared in quadruplicate. After incubation, RCAs were stopped by adding 250 µL of HPLC-grade methanol (Lach-Ner, Czech Republic) to each tube. Samples were vortexed and stored at −20°C. Before HPLC analysis, samples were briefly vortexed, sonicated for 5 min, again briefly vortexed, and 250 µL of the solution was transferred to 1.5 mL microtubes. These were centrifuged at maximum speed for 3 min, and 150 µL of the supernatant was transferred into HPLC vials with glass inserts and stored at −20°C until analysis. The HPLC analysis was performed according to (45) with modifications described in the Supplemental Material.

## Taxonomic composition of the ASC12, sequencing, and assembly of the ASC12 metagenome

Metagenomic DNA was isolated from the cell pellets after the centrifugation of 100 µL of bacterial cell suspension for 10 min at 5,000 × $g$ by PureLink Genomic DNA Mini Kit according to the manufacturer's instructions. The taxonomic composition of the ASC12 consortium was analyzed by 16S rRNA gene amplicon sequencing following the procedure described by Papik et al. (46). Shotgun metagenomic sequencing was performed using both Illumina iSeq 100 and Oxford Nanopore GridIONx5, with quality control and assembly of the sequencing data performed as described in the Supplemental Material.

## Screening MAGs for AS and CP-transforming enzymes

A local gene database, designated ASD_DB, was constructed to include protein sequences of genes with a literature-confirmed role in bacterial catabolism of S-lignin-derived compounds and CPs, as well as their homologous sequences. A protein catalog was generated from the annotated coding regions of assembled MAGs. This catalog was then used for bulk screening of proteins of interest.

Protein sequences from the annotated MAGs were compared against the ASD_DB using BLASTP v.2.16.0 (47). Sequences that shared at least >40% identity with entries in ASD_DB were manually annotated using InterPro. If the annotations suggested potential involvement in the transformation of AS or CPs, these sequences were labeled as "The most promising hits." This subset of proteins was subsequently subjected to experimental validation to confirm their role in the transformation of AS, 2,4,6-TCP, and 2,6-DCP.

## Isolation of a bacterial pure culture

An aliquot of the enriched bacterial consortium ASC12 was serially diluted, and 100 µL of each dilution was transferred in duplicates to a plate containing R2A medium (Difco, UK). After 72 h of cultivation at 12°C, individual colonies were isolated, dereplicated, and identified using the MALDI-TOF MS with the mMass software (48) and the MALDI BioTyper 3.1 (Bruker, Germany) database, respectively. Both identified and yet unidentified colonies were subjected to Sanger sequencing (49) of the nearly full 16S rRNA gene using 8F (AGAGTTTGATCMTGGCTCAG) and 1509R (GYTACCTTGTTACGACTT) primers.

## 2,6-DCP and 2,4,6-TCP transformation by *Pseudomonas rhizophila* AS1

Cells of the isolated *Pseudomonas rhizophila* pure culture designated AS1 were prepared similarly to the ASC12 for the RCAs. Briefly, AS1 cell suspensions were washed and adjusted to $OD_{600} = 1$ in MM, then split into active and autoclaved (negative control) portions. For RCAs, 250 µL of cell suspension was incubated with 2.5 µL of 10 mM 2,6-DCP and 2,4,6-TCP stock solutions at 12°C for 48 h with shaking. Reactions were stopped with methanol, processed by vortexing, sonication, and centrifugation, and supernatants were transferred to HPLC vials for subsequent HPLC-PDA analysis. All samples were prepared in triplicate.

## Heterologous expression of AS1 genes in *Escherichia coli*

To identify the genetic determinant of AS transformation, a series of pET-19b-based plasmids (Novagen) bearing individual candidate genes was prepared. Briefly, the pET-19b backbone and coding candidate sequences were PCR-amplified using primers indicated in Table S1, fused using the NEBuilder HiFi DNA Assembly Cloning Kit (New England Biolabs) according to the manufacturer's recommendations, and transformed into *E. coli* DH5α cells. Transformants were selected on solid LB plates amended with ampicillin (150 µg/mL). The DNA sequence of the inserts was then verified by Sanger sequencing. The expression of candidate genes was performed according to (41, 50) with modifications described in the Supplemental Material.

## Phylogenetic classification of the *asdA* gene

AsdA protein sequence was queried against the NCBI nr-protein database (51) using BLASTP (52), and protein sequences of the first 100 hits with the lowest E-value were downloaded. The sequence set was aligned by MAFFT v.7.526 (53) with the L-INS-I strategy. The resulting multiple sequence alignment was trimmed by ClipKIT v.2.4.1 (54) in smart-gap mode. The phylogeny was reconstructed with IQ-TREE v.2.4.0 (55). Best-fit model LG+F+I+R5 was chosen according to BIC using ModelFinder (55). Branch support of the phylogenetic tree was tested with UFBoot (56).

To assess the uniqueness of the *asdA* gene region, also referred to as the *asdBCADE* gene cluster, we used Clinker (57) to compare it with gene clusters of representative strains carrying the closest homologs of *asdA*. The gene cluster of *Pseudomonas putida* WBC-3 was included in the analysis because its genome contains the *pnpA* gene, which was part of the ASD_DB, and through its homology, the *asdA* gene was found using BLAST.

## Preparation of the AS1-derived biosensor strain and the induction assay

To investigate the ability of AS and AS-like compounds to induce enzymes encoded by the putative *asd* operon, a biosensor strain derived from AS1 was constructed. For this purpose, the pTr-IR-egfp plasmid, a shuttle vector based on the *E. coli-Pseudomonas* plasmid pUCP-18, was used, as described in Suman et al. (50). The plasmid contains a 1,270 bp intergenic region (IR) upstream of the *asd* operon, positioned adjacent to the *egfp* (green fluorescent protein) reporter gene. Fluorescence from eGFP was measured as a proxy for *asd* gene expression in response to the substrates. The preparation of the AS1-derived biosensor strain, herein referred to as AS1/eGFP, and the induction assay are described in the Supplemental Material.

## Analysis of AS and 2,4,6-TCP transformation products

Enzyme activity of *asdA*- and *asdBCAD*-bearing *E. coli* BL21(DE3) cells was assessed using a procedure similar to the RCA described above. Briefly, 250 µL of the cell suspension ($OD_{600}$ = 10) was added to 4 mL glass vials along with 2.5 µL of 10 mM stock solutions of AS or 2,4,6-TCP. Biotic controls consisted of *E. coli* BL21(DE3) without any plasmid insert, and abiotic controls contained no cells. All reactions were prepared in quadruplicate. The incubations were carried out for 24 h at 28°C and 150 rpm. To stop the reactions, 250 µL of HPLC-grade methanol was added to each vial. Before LC-MS analysis, cells were removed by centrifugation at maximum speed for 3 min, and the supernatant was transferred to fresh glass vials for LC-MS analysis.

Analysis of AS and 2,4,6-TCP transformation products in the extract was performed using an Agilent 1260 Infinity II liquid chromatograph coupled to a high-resolution Agilent QTOF 6546 mass spectrometer–QTOF MS, equipped with an Agilent Jet Stream electrospray ion source (Agilent Technologies, CA, USA) as described in the Supplemental Material.

## Distribution of *Pseudomonas rhizophila* and *asdA* across habitats

For profiling of species-representative genomes belonging to the family *Pseudomonadaceae* according to GTDB r220 (58), a local protein database of 1,007 genomes was created with doggo_fetch from WhereDoGGo? pipeline (https://github.com/MEDEAlab/WhereDoGGo) leveraging ncbi-datasets-cli 16.36.0 (https://github.com/ncbi/datasets) for downloading genomes and Pyrodigal 3.5.1 (59) for ORF prediction. The profiling of all the genomes belonging to the genus *Pseudomonas_E* was performed in a similar manner, except that 7,438 genomes were downloaded using ncbi-datasets-cli, and a local protein database was created with doggo_herd from WhereDoGGo?. AsdA sequence was used as a query against these databases using diamond BLASTP (60).

Metagenome profiling was conducted by downloading up to 10 Gb of whole metagenomic data using sra-tools release 3.1.1 (https://github.com/ncbi/sra-tools). The

metagenomes were selected by searching for s__*Pseudomonas_E rhizophila* in the Sandpiper database (61). A single-protein DIAMOND database was built from the AsdA sequence (diamond makedb), and reads were queried with diamond blastx. To increase specificity, only reads mapping between the two conserved FAD/NADH-binding domains of AsdA (residues 183–265) with an E-value <0.01 were retained.

## Assessing the role of AS1 and *asdA* in the S-lignin building-blocks catabolism

To characterize the catabolic potential of *Pseudomonas rhizophila* AS1, RCAs were conducted to assess the transformation of selected aromatic compounds associated with S-lignin metabolism. Briefly, AS1 cells were precultivated in MM with 1 mM of AS and harvested in the late exponential phase. After washing and resuspension in sterile MM to an $OD_{600}$ of 1, 250 µL of the active and autoclaved (negative control) suspension was added to sterile 4 mL glass tubes containing 2.5 µL of individual 10 mM stock solutions of selected aromatic compounds representing S-lignin building blocks. Reactions were incubated for 48 h at 12°C with shaking at 100 rpm. Each condition was tested in triplicate. Reactions were stopped by the addition of 250 µL of HPLC-grade methanol, followed by vortexing and centrifugation at maximum speed for 3 min. The supernatants were transferred to fresh glass vials and stored at −20°C until HPLC-PDA analysis.

To evaluate the functional role of the *asdA* gene, the same compounds were tested using induced *E. coli* BL21(DE3) expressing either *asdA* or an empty vector control. Cells were prepared and assayed as described above, with incubation carried out for 24 h at 28°C and 150 rpm. Substrate depletion was monitored by HPLC-PDA, and depletion was quantified by comparing residual substrate concentrations to control samples.

## ACKNOWLEDGMENTS

This work was funded by a grant from the Czech Science Foundation (project no. 22-00132S) and by a grant from the Programme Johannes Amos Comenius under the Ministry of Education, Youth and Sports of the Czech Republic (project no. CZ.02.01.01/00/22_008/0004597).

Conceptualization - O.U., M.S., J.S., T.E., L.J.; Data curation - T.E., L.J., Z.S., S.C., P.P., M.C., M.S., J.S.; Formal analysis - T.E., Z.S., S.C., R.S., T.C., M.S., J.S.; Funding acquisition & resources - O.U., T.C., P.P.; Investigation - T.E., L.J., Z.S., S.C., M.F., M.C., J.S.; Methodology - O.U., M.S., J.S., T.C., T.E., P.P.; Supervision - O.U.; Validation - O.U., M.S., J.S., T.C., T.E.; Visualization - T.E., M.S.; Writing – original draft - T.E.; Writing – review & editing - O.U. and all other co-authors.

## AUTHOR AFFILIATIONS

[1]Department of Biochemistry and Microbiology, Faculty of Food and Biochemical Technology, University of Chemistry and Technology, Prague, Czech Republic
[2]Institute of Microbiology, Academy of Sciences of the Czech Republic, Prague, Czech Republic
[3]Military Health Institute, Military Medical Agency, Prague, Czech Republic
[4]Department of Infectious Diseases, First Faculty of Medicine, Charles University and Military University Hospital Prague, Prague, Czech Republic

## AUTHOR ORCIDs

Tomas Engl http://orcid.org/0009-0005-1594-3102
Lydie Jakubova http://orcid.org/0009-0006-6361-831X
Zdena Skrob http://orcid.org/0000-0002-4140-1332
Stephanie Campeggi http://orcid.org/0009-0006-6278-6952
Roman Skala http://orcid.org/0000-0002-0549-6709
Magdalena Folkmanova http://orcid.org/0000-0002-6106-5137
Petr Pajer http://orcid.org/0000-0002-8706-9371

Martin Chmel 🄳 http://orcid.org/0000-0003-0864-0574
Tomas Cajthaml 🄳 http://orcid.org/0000-0002-3393-1333
Michal Strejcek 🄳 http://orcid.org/0000-0002-6755-5356
Jachym Suman 🄳 http://orcid.org/0000-0003-1828-8391
Ondrej Uhlik 🄳 http://orcid.org/0000-0002-0506-202X

## FUNDING

| Funder | Grant(s) | Author(s) |
| --- | --- | --- |
| Grantová Agentura České Republiky | 22-00132S | Tomas Engl |
| | | Lydie Jakubova |
| | | Michal Strejcek |
| | | Jachym Suman |
| | | Ondrej Uhlik |
| Ministerstvo Školství, Mládeže a Tělovýchovy | CZ.02.01.01/00/22_008/0004597 | Tomas Engl |
| | | Stephanie Campeggi |
| | | Roman Skala |
| | | Michal Strejcek |
| | | Jachym Suman |
| | | Ondrej Uhlik |

## DATA AVAILABILITY

All raw sequencing data generated in this study are available in the GenBank database under the BioProject accession number PRJNA1219158. The measured analytical data and complete ASD_DB database are available online at https://doi.org/10.5281/zenodo.15525860. The authors confirm that all supporting data and protocols are provided within the article or through the Supplemental Material files.

## ADDITIONAL FILES

The following material is available online.

### Supplemental Material

**Supplemental Material (mSystems01242-25-S0001.pdf).** Supplemental methods, figures, and tables.

### Open Peer Review

**PEER REVIEW HISTORY (review-history.pdf).** An accounting of the reviewer comments and feedback.

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
