## [Reviewer comments · mSystems]

Catabolism of Acetosyringone and Co-metabolic Transformation of 2,4,6-Trichlorophenol by a Novel FAD-dependent Monooxygenase

Tomas Engl, Lydie Jakubova, Zdena Skrob, Stephanie Campeggi, Roman Skala, Magdalena Folkmanova, Petr Pajer, Martin Chmel, Tomáš Cajthaml, Michal Strejcek, Jachym Suman, and Ondrej Uhlik

Corresponding Author(s): Ondrej Uhlik, University of Chemistry and Technology, Prague

Review Timeline:

Submission Date:	August 21, 2025
Editorial Decision:	October 20, 2025
Revision Received:	November 19, 2025
Editorial Decision:	January 5, 2026
Revision Received:	January 5, 2026
Accepted:	January 7, 2026

Editor: Lennart Schada von Borzyskowski

Reviewer(s): Disclosure of reviewer identity is with reference to reviewer comments included in decision letter(s). The following individuals involved in review of your submission have agreed to reveal their identity: Celso Martins (Reviewer #1)

Transaction Report:

DOI: <https://doi.org/10.1128/mSystems.01242-25>

Re: mSystems01242-25 (Catabolism of Acetosyringone and Co-metabolic Degradation of 2,4,6-Trichlorophenol by a Novel FAD-dependent Monooxygenase)

Dear Prof. Ondrej Uhlík:

Your manuscript has now been seen by three expert reviewers. While all reviewers acknowledge that your study is of good quality and reports interesting novel results, all three reviewers also suggest important points that require improvement. In a revised version of your manuscript, please address all reviewer comments.

Please ensure to pay special attention to:

- the question whether TCP is detoxified or fully degraded. To demonstrate complete biodegradation of TCP, additional experiments would be required, as suggested by the reviewers. In contrast, the detoxification of TCP is already well supported by the currently available data.
- the fact that some parts of the Methods section should be explained in the Results section, as pointed out by Reviewer 3

Revision Guidelines

Sincerely,

Reviewer #1 (Comments for the Author):

The manuscript "Catabolism of Acetosyringone and Co-metabolic Degradation of 2,4,6-Trichlorophenol by a Novel FAD-dependent Monooxygenase" by T. Engl et al. presents a novel metabolic pathway for the degradation of the lignin-derived acetosyringone, and shows links between the degradation of this natural polymer and man-made chlorophenol pollutants, in a bacterial consortium.

The study is overall well-written and presents interesting data, as well as sound experimental design and methodologies. There are, however, a few points this reviewer would like to see clarified before recommending for publication in mSystems.

Major concern to be revised:

Throughout the manuscript the authors claim degradation of chlorinated phenols. However, The transformation of chlorophenols into hydroquinones, catechols, and related compounds has been extensively described in the past, primarily using either filamentous fungi or *Sphingobium chlorophenolicum*. Such additional hydroxylations (particularly at the para and meta positions) are recognized as solubilization steps, enhancing the bioavailability of the toxins for subsequent transformations. In contrast, O-methylations, which are phase II conjugations, usually occur in already hydroxylated transformation products and represent particularly challenging steps for the organism, as they involve processing both the parent compound and its phase I derivative (in this case, hydroquinone).

Taken together, these observations - along with the absence of transformation products downstream of the dichlorinated variants, suggest that the initial phenols are being modified primarily to reduce environmental toxicity - rather than being fully degraded (which would imply their utilization as a carbon source).

I would be interested to know whether the authors detected metabolites that could indicate channeling into catabolic pathways (i.e., non-chlorinated compounds, particularly those related to the TCA cycle, even if at slightly upstream points). However, given the co-metabolic nature of these transformation steps, it would be virtually impossible to demonstrate that such central metabolites originated specifically from chlorophenol degradation, rather than from the catabolism of AS, which the authors convincingly demonstrated to be used as a sole carbon source.

In this context, I believe it would be more accurate for the authors to revise the claims of "chlorophenol degradation" to "chlorophenol transformation," since no definitive proof of degradation was provided.

If the authors wish to pursue the degradation hypothesis, I would recommend using isotopically labeled chlorophenols to trace their transformation products via metabolomics. Nevertheless, I consider the study highly valuable in its current form, as it highlights the potential role of the studied species/consortium in contributing to chlorophenol degradation in natural environments, even if this occurs in cooperation with other organisms that are more efficient at carrying out the transformations required for complete pollutant mineralization.

Minor concerns:

Line 72: Please define Acetosyringone (AS) the first time it appears in the main text.

Line 104: Why resting cells? Maybe degradation levels get improved if you try orbital agitation, which will increase the redox potential of the solution due to ion exchange.

Line 204: It is not clear to me how the authors could discriminate which position was hydroxylated. Could you please clarify? Was the fragmentation analysis enough to prove this detail? The same for other compounds described later in the text.

Reviewer #2 (Comments for the Author):

The manuscript by Engl et al. aims to demonstrate the existence of a novel metabolic pathway for acetosyringone (AS) catabolism, initiated by a previously uncharacterized FAD-dependent monooxygenase (*AsdA*), which additionally catalyzes the co-metabolic degradation of 2,4,6-TCP. The work presents a genuinely novel discovery: an AS degradation route that, unlike the only previously documented mechanism (side chain modification in *Sphingobium* sp. SYK-6), operates through direct aromatic ring hydroxylation. The methodological strategy is robust: enrichment with AS as sole carbon source, metagenomic analysis of ASC12 consortium, isolation of *Pseudomonas rhizophila* AS1, heterologous expression of candidate genes, and LC-MS/MS characterization of transformation products. The identification of AS-OH (C₁₀H₁₂O₅) and DCHQ as metabolites unequivocally confirms *AsdA* activity. However the authors do NOT demonstrate that the AS-OH product is effectively metabolized by the complete strain to integration into central metabolism. They only show that cells harboring the complete *asdBCAD* cluster do not accumulate AS-OH, but they fail to identify final products or subsequent intermediates. Therefore, it is unclear whether *AsdA* activity constitutes the first step of a complete catabolic pathway or represents a co-metabolic or detoxification reaction. The authors should consider rephrasing this claim or explicitly addressing this limitation in the discussion.

Other issues to be considered,

-The link between the enriched consortium (ASC12) and the isolated strain (*Pseudomonas rhizophila* AS1) remains somewhat ambiguous. Although both exhibit activity toward 2,4,6-TCP, the manuscript does not clearly establish whether the consortium's degradation capacity is solely attributable to AS1 or to potential synergistic interactions with *Methylotenera* or other community members. This experimental transition should be clarified and possible co-metabolic effects discussed at the consortium level.

-The case of orcinol is particularly intriguing: it induces expression of the *asd* operon in the biosensor strain, yet AS1 does not grow on or degrade it. This uncoupling between regulatory induction and catalytic activity suggests possible regulatory promiscuity or recognition of structural analogs of acetosyringone. Consideration should be given to discussing this observation and its implications in the discussion section

-The phylogenetic analysis reveals that *AsdA* represents a distinct evolutionary branch within FAD-dependent monooxygenases, suggesting divergent evolution from a common ancestor. The authors could briefly discuss *AsdA* as a potential example of functional diversification, where catalytic promiscuity enables the transformation of both lignocellulosic metabolites (acetosyringone) and structurally analogous chlorinated pollutants (2,4,6-TCP). This highlights the enzyme's evolutionary adaptability within its protein family

Reviewer #3 (Comments for the Author):

Engl et al. used an enrichment culture to isolate a mixed bacterial community capable of degrading acetosyringone. They then isolate a single *Pseudomonas* strain that can grow with acetosyringone as a sole carbon source, and demonstrate that this strain also co-metabolizes 2,4,6-trichlorophenol. They identify the first gene in the catabolic pathway, a monooxygenase they name *asdA*, and characterize its substrate range and metagenomic prevalence. In general, I thought the study was well-done. I had concerns about the level of detail provided in the manuscript, and was unconvinced by the analysis of the distribution of *asdA*. There were things I would have liked to see included (more characterization of *asdA*, such as purification and enzyme kinetics; more attention given to the whole pathway) but I don't think they're strictly necessary.

Major comments:

1. The enrichment experiment is, in my opinion, properly part of the results not just the methods. Please provide a brief summary of the experiment before jumping to the 'ASC12' designation. Were there changes in the growth of the mixed culture during the enrichment? E.g. the methods say that the culture was diluted every 1-2 weeks. Was that initially 2 weeks and then shortened to 1 week? How did you decide when to passage the culture?
2. I would appreciate more details on the ASD_DB. What genes were included in the search? An SI table would be appropriate. Similarly, the development of this database, unless previously described, should be explained in the results and discussion not just the methods.
3. Same comment on line 188, the design and construction of pTr-IR-egfp merits more than a single mention of 'the plasmid was transformed'
4. And similar comments on line 195. Please briefly describe the experiment before jumping into the strains and analytical methods.
5. Line 221 - the 1007 genomes were screened for what, exactly? *asdA*?
6. Line 228 - *asdA* was found in 9 metagenomes, 'primarily from rhizosphere'. What proportion of the metagenomes were rhizosphere, and is that higher than in the original database? I'm not convinced that the data demonstrate that *asdA* is truly enriched in 'soils enriched in organic matter'.
7. Line 282 - Has it been demonstrated that MT1 enhances AS degradation? As noted, I couldn't find evidence that the mixed culture is more efficient than the pure culture of AS1.
8. I appreciate the decision that a complete analysis of AS degradation by AS1 is out of scope. However, it was disappointing to not even see a hypothesis about how those genes could assemble into a pathway.

Minor comments:

1. Strain SYK-6 is now formally *Sphingobium lignivorans* SYK-6. It's also odd to cite reference 8 to discuss AS catabolism or SYK-6 catabolic pathways.
2. It's a little odd that ASC12 was not tested with AS as a substrate. Is the AS fully degraded? And how does the degradation of the full mixed culture compare to degradation by AS1?
3. What does the acronym 'RCA' stand for? I can't find it defined anywhere.
4. Line 184 - 'steps of AS and 2,4,6-TCP'.
5. Figure 9 - what does 'relative depletion' measure? Relative to the initial concentration?

Response to Reviewers

We would like to thank the editor and the three reviewers for their constructive and insightful comments. We have revised the manuscript accordingly. Our responses to the comments are provided below.

Reviewer #1 (Comments for the Author):

The manuscript "Catabolism of Acetosyringone and Co-metabolic Degradation of 2,4,6-Trichlorophenol by a Novel FAD-dependent Monooxygenase" by T. Engl et al. presents a novel metabolic pathway for the degradation of the lignin-derived acetosyringone, and shows links between the degradation of this natural polymer and man-made chlorophenol pollutants, in a bacterial consortium.

The study is overall well-written and presents interesting data, as well as sound experimental design and methodologies. There are, however, a few points this reviewer would like to see clarified before recommending for publication in mSystems.

Major concern to be revised:

Throughout the manuscript, the authors claim degradation of chlorinated phenols. However, the transformation of chlorophenols into hydroquinones, catechols, and related compounds has been extensively described in the past, primarily using either filamentous fungi or *Sphingobium chlorophenolicum*. Such additional hydroxylations (particularly at the para and meta positions) are recognized as solubilization steps, enhancing the bioavailability of the toxins for subsequent transformations. In contrast, O-methylations, which are phase II conjugations, usually occur in already hydroxylated transformation products and represent particularly challenging steps for the organism, as they involve processing both the parent compound and its phase I derivative (in this case, hydroquinone).

Taken together, these observations - along with the absence of transformation products downstream of the dichlorinated variants, suggest that the initial phenols are being modified primarily to reduce environmental toxicity - rather than being fully degraded (which would imply their utilization as a carbon source).

I would be interested to know whether the authors detected metabolites that could indicate channeling into catabolic pathways (i.e., non-chlorinated compounds, particularly those related to the TCA cycle, even if at slightly upstream points). However, given the co-metabolic nature of these transformation steps, it would be virtually impossible to demonstrate that such central metabolites originated specifically from chlorophenol degradation, rather than from the catabolism of AS, which the authors convincingly demonstrated to be used as a sole carbon source.

In this context, I believe it would be more accurate for the authors to revise the claims of "chlorophenol degradation" to "chlorophenol transformation," since no definitive proof of degradation was provided.

If the authors wish to pursue the degradation hypothesis, I would recommend using isotopically

labeled chlorophenols to trace their transformation products via metabolomics. Nevertheless, I consider the study highly valuable in its current form, as it highlights the potential role of the studied species/consortium in contributing to chlorophenol degradation in natural environments, even if this occurs in cooperation with other organisms that are more efficient at carrying out the transformations required for complete pollutant mineralization.

- We agree that our data for 2,4,6-TCP and 2,6-DCP support co-metabolic transformation rather than complete mineralization. We therefore systematically replaced the terms “*degradation*” or “*breakdown*” of CPs or AS by “*transformation*” throughout the manuscript. This includes the Abstract, Introduction, Results (sections on APs/CPs and on AsdA activity), Figure captions, and the Discussion. We now explicitly state that we demonstrate the initial step of 2,4,6-CPs transformation into DCHQ and formation of DCMP, but do not resolve further catabolic steps or integration into central metabolism.
- We also agree that our evidence supports the putative role of the *asdB**BCAD* genes in the downstream transformation of AS-OH, but it does not delineate a complete pathway. In the Discussion of the revised manuscript, we have revised our wording and now describe the *asdB**BCAD* cluster as “a candidate operon likely involved in further transformation of the AS-OH”. We explicitly acknowledge that we did not detect TCA-cycle intermediates attributable to AS and that the complete pathway remains unresolved.

Minor concerns:

Line 72: Please define Acetosyringone (AS) the first time it appears in the main text.

- Modified as requested.

Line 104: Why resting cells? Maybe degradation levels get improved if you try orbital agitation, which will increase the redox potential of the solution due to ion exchange.

- Our aim was not to optimize conditions for maximal pollutant removal but rather to compare the transforming potential of different strains and constructs under a controlled, reproducible setup. Resting-cell assay is a format we have previously used and validated for this purpose. It allows us to (i) decouple transformation activity from growth, and (ii) maintain a defined physiological state induced by pre-growth on AS.

Line 204: It is not clear to me how the authors could discriminate which position was hydroxylated. Could you please clarify? Was the fragmentation analysis enough to prove this detail? The same for other compounds described later in the text.

- The assignment of the ortho-hydroxylation of AS was based on high-resolution MS/MS fragmentation of AS-OH and the corresponding isotopic pattern. The observed fragment

ions and their neutral losses are only consistent with hydroxylation on the aromatic ring at the 2,6-position, rather than on the side chain or at other ring positions (Line 212).

Reviewer #2 (Comments for the Author):

The manuscript by Engl et al. aims to demonstrate the existence of a novel metabolic pathway for acetosyringone (AS) catabolism, initiated by a previously uncharacterized FAD-dependent monooxygenase (AsdA), which additionally catalyzes the co-metabolic degradation of 2,4,6-TCP. The work presents a genuinely novel discovery: an AS degradation route that, unlike the only previously documented mechanism (side chain modification in *Sphingobium* sp. SYK-6), operates through direct aromatic ring hydroxylation. The methodological strategy is robust: enrichment with AS as sole carbon source, metagenomic analysis of ASC12 consortium, isolation of *Pseudomonas rhizophila* AS1, heterologous expression of candidate genes, and LC-MS/MS characterization of transformation products. The identification of AS-OH (C₁₀H₁₂O₅) and DCHQ as metabolites unequivocally confirms AsdA activity. However the authors do NOT demonstrate that the AS-OH product is effectively metabolized by the complete strain to integration into central metabolism. They only show that cells harboring the complete asdBCAD cluster do not accumulate AS-OH, but they fail to identify final products or subsequent intermediates. Therefore, it is unclear whether AsdA activity constitutes the first step of a complete catabolic pathway or represents a co-metabolic or detoxification reaction. The authors should consider rephrasing this claim or explicitly addressing this limitation in the discussion.

- We agree that our current data do not directly trace AS-derived carbon from AS-OH into central metabolism. We have therefore rephrased our claims. We now state that AsdA initiates an AS transformation route through direct aromatic ring hydroxylation and that *P. rhizophila* AS1 grows on AS as its sole carbon source. While this implies the existence of a complete catabolic pathway, we acknowledge that only the first step has undergone functional characterization.

Other issues to be considered,

The link between the enriched consortium (ASC12) and the isolated strain (*Pseudomonas rhizophila* AS1) remains somewhat ambiguous. Although both exhibit activity toward 2,4,6-TCP, the manuscript does not clearly establish whether the consortium's degradation capacity is solely attributable to AS1 or to potential synergistic interactions with *Methylotenera* or other community members. This experimental transition should be clarified and possible co-metabolic effects discussed at the consortium level.

- The AS1 isolate exhibited high and specific depletion of 2,6-DCP and 2,4,6-TCP, comparable to that of the ASC12 consortium. When comparing the depletion of these two chlorophenols as well as AS, we did not observe any significant difference between AS1 and the ASC12 consortium (Line 148-154).

The case of orcinol is particularly intriguing: it induces expression of the *asd* operon in the biosensor strain, yet AS1 does not grow on or degrade it. This uncoupling between regulatory induction and catalytic activity suggests possible regulatory promiscuity or recognition of structural analogs of acetosyringone. Consideration should be given to discussing this observation and its implications in the discussion section

- We agree with the reviewer that the phenomenon described is particularly intriguing. To address this, we have revised the Discussion section (line 293 and further).

The phylogenetic analysis reveals that *AsdA* represents a distinct evolutionary branch within FAD-dependent monooxygenases, suggesting divergent evolution from a common ancestor. The authors could briefly discuss *AsdA* as a potential example of functional diversification, where catalytic promiscuity enables the transformation of both lignocellulosic metabolites (acetosyringone) and structurally analogous chlorinated pollutants (2,4,6-TCP). This highlights the enzyme's evolutionary adaptability within its protein family

- This issue was addressed in the Discussion section. As *AsdA* is a phylogenetically divergent enzyme, it is consistent that we identified a previously undescribed mechanism for the transformation of AS to AS-OH. The transformation of 2,4,6-TCP appears to result from both substrate promiscuity of the enzyme and a cometabolic effect, as AS1 does not grow on 2,4,6-TCP and is able to transform it only in the presence of an inducer (AS). All of these aspects are described in the manuscript. However, fully explaining how and why the enzyme diverged functionally from other members of the same family would require additional phylogenetic and evolutionary analyses.

Reviewer #3 (Comments for the Author):

Engl et al. used an enrichment culture to isolate a mixed bacterial community capable of degrading acetosyringone. They then isolate a single *Pseudomonas* strain that can grow with acetosyringone as a sole carbon source, and demonstrate that this strain also co-metabolizes 2,4,6-trichlorophenol. They identify the first gene in the catabolic pathway, a monooxygenase they name *asdA*, and characterize its substrate range and metagenomic prevalence. In general, I thought the study was well-done. I had concerns about the level of detail provided in the manuscript, and was unconvinced by the analysis of the distribution of *asdA*. There were things I would have liked to see included (more characterization of *asdA*, such as purification and enzyme kinetics; more attention given to the whole pathway) but I don't think they're strictly necessary.

Major comments:

1. The enrichment experiment is, in my opinion, properly part of the results not just the methods. Please provide a brief summary of the experiment before jumping to the 'ASC12' designation. Were there changes in the growth of the mixed culture during the enrichment? E.g. the methods say that the culture was diluted every 1-2 weeks. Was that initially 2 weeks and then shortened to 1 week? How did you decide when to passage the culture?

- We agree that a brief description in the Results improves readability. We have added a paragraph at the beginning of the Results section.

2. I would appreciate more details on the ASD_DB. What genes were included in the search? An SI table would be appropriate. Similarly, the development of this database, unless previously described, should be explained in the results and discussion not just the methods.

- All the information on this database, including what genes were included in the search, is available at <https://doi.org/10.5281/zenodo.15525860> as specified on line 559.

3. Same comment on line 188, the design and construction of pTr-IR-egfp merits more than a single mention of 'the plasmid was transformed'

- Additional information regarding the preparation of the biosensor strain is available in the Supplementary material, section “The preparation of the AS1-derived biosensor strain and the induction assay”.

4. And similar comments on line 195. Please briefly describe the experiment before jumping into the strains and analytical methods.

- Detailed description of the procedure of the experiment is provided in the Supplementary material, section “The LC-MS analysis specifications”. The decision to include details on the methods in the Supplementary material was driven by the mSystems journal's overall character limit for papers.

5. Line 221 - the 1007 genomes were screened for what, exactly? asdA?

- Yes, we have corrected this section (Line 210).

6. Line 228 - asdA was found in 9 metagenomes, 'primarily from rhizosphere'. What proportion of the metagenomes were rhizosphere, and is that higher than in the original database? I'm not convinced that the data demonstrate that asdA is truly enriched in 'soils enriched in organic matter'.

- We agree that our dataset does not support a strong claim of enrichment. Therefore, we have revised the language. We now state:

“Within the examined metagenomes, *asdA*-positive reads were primarily detected in rhizosphere and compost samples, i.e. in plant-associated or organic-matter-rich environments. However, the small number of hits and biased sampling prevent a rigorous statistical assessment of habitat enrichment.”

7. Line 282 - Has it been demonstrated that MT1 enhances AS degradation? As noted, I couldn't find evidence that the mixed culture is more efficient than the pure culture of AS1.

- ASC12 was indeed tested on AS and showed complete AS depletion under the conditions used for RCAs. We did not observe a clear advantage of ASC12 over AS1 in terms of AS depletion under our conditions (data not presented in this manuscript). Therefore, the role of detoxification of C1 compounds as formaldehyde or methanol, and enhancing the AS depletion by the MT1 remains solely hypothetical.

8. I appreciate the decision that a complete analysis of AS degradation by AS1 is out of scope. However, it was disappointing to not even see a hypothesis about how those genes could assemble into a pathway.

- Given that the *asd* gene cluster contains two α/β -hydrolases (AsdB and AsdD) for which no functional information is available, it is also challenging to predict the intermediates formed during the transformation of AS-OH. The VOC-family protein ACNJA3_12685 (AsdC) is expected to function as a cleaving enzyme within the *asd* gene cluster. However, extensive experimental work will be required to characterise the entire cluster and to determine the complete pathway.

Minor comments:

1. Strain SYK-6 is now formally *Sphingobium lignivorans* SYK-6. It's also odd to cite reference 8 to discuss AS catabolism or SYK-6 catabolic pathways.

- We have updated the taxonomy of *Sphingobium sp.* SYK6 to *Sphingobium lignivorans* SYK6 and used different sources for citing SYK-6 catabolic pathways.

2. It's a little odd that ASC12 was not tested with AS as a substrate. Is the AS fully degraded? And how does the degradation of the full mixed culture compare to degradation by AS1?

- ASC12 was tested with AS as a substrate. Although the corresponding results are not shown, we observed complete depletion of AS. However, a comprehensive non-targeted metabolomic analysis was not performed, as such data were beyond the scope of this study. The growth curves of the ASC12 consortium and the AS1 strain did not differ significantly when AS was used as a sole carbon source. Nevertheless, since metabolomic data are lacking, the C1 compound metabolism by MT1 within the ASC12 remains hypothetical.

3. What does the acronym 'RCA' stand for? I can't find it defined anywhere.

- We have defined the RCA correctly.

4. Line 184 (164) - 'steps of AS and 2,4,6-TCP'.

- We have corrected this mistake.

5. Figure 9 - what does 'relative depletion' measure? Relative to the initial concentration?

- Relative depletion denotes the percentage decrease in substrate peak area relative to the corresponding control, normalised to the initial substrate concentration.

Re: mSystems01242-25R1 (Catabolism of Acetosyringone and Co-metabolic Transformation of 2,4,6-Trichlorophenol by a Novel FAD-dependent Monooxygenase)

Dear Prof. Ondrej Uhlík:

Your revised manuscript was sent to three reviewers again. While reviewers 1 and 3 do not have any further comments, reviewer 2 raises one additional point as well as a few minor comments. Please check these comments and address them in a final, revised version of your manuscript.

Revision Guidelines

Sincerely,
Lennart Schada von Borzyskowski
Editor
mSystems

Reviewer #2 (Comments for the Author):

This is an improved version of the previous submitted manuscript. The main concern about the absence of evidence to claim a novel metabolic pathway has been addressed. However there is a small but significant detail that should be clarified.

In lines 259-262 it is stated,

"Strain AS1 depleted several compounds, including the previously mentioned AS, as well as several structurally related compounds: caffeic acid, p-coumaric acid, ferulic acid, 2,4-dihydroxybenzoic acid, SYRINGIC ACID, vanillic acid, and veratric acid"; however in Figure 9 syringic acid is absent. It has been tested syringic acid (SA) as a substrate? This is very relevant since the previous reported pathway for AS in *Sphingobium lignovorans* SYK-6 involves SA as the key intermediate. Briefly, depletion of SA should be evaluated in *P. rhizophila* AS1 precultivated with AS as the sole carbon source versus an alternative substrate to check whether SA consumption is induced by AS catabolism.

Minor comments

lines 326-330 ...beyond the scope of this study/work is redundant in the same paragraph

line 511 Pyrodigal should be changed to Prodigal

Accession numbers of assembled MAGs should be included in the main manuscript

Reviewer #3 (Comments for the Author):

My concerns have been addressed to my satisfaction.

Response to Reviewer 2's comments

We have revised the manuscript according to the comments provided. Our responses to each comment are provided below.

Reviewer #2 (Comments for the Author):

This is a improved version of the previous submitted manuscript. The main concern about the absence of evidence to claim a novel metabolic pathway has been addressed. However there a small but significant detail that should be clarified.

In lines 259-262 it is stated,

"Strain AS1 depleted several compounds, including the previously mentioned AS, as well as several structurally related compounds: caffeic acid, p-coumaric acid, ferulic acid, 2,4-dihydroxybenzoic acid, SYRINGIC ACID, vanillic acid, and veratric acid"; however in Figure 9 syringic acid is absent.

It has been tested syringic acid (SA) as a substrate? This is very relevant since the previous reported pathway for AS in *Sphingobium lignovorans* SYK-6 involves SA as the key intermediate. Briefly, depletion of SA should be evaluated in *P. rhizophila* AS1 precultivated with AS as the sole carbon source versus an alternative substrate to check whether SA consumption is induced by AS catabolism.

We thank the reviewer for pointing out an error in the text. Indeed, it should say "sinapic acid" instead of "syringic acid." As indicated in Fig. 9, sinapic acid, but not syringic acid, was tested as a substrate of AS1/AsdA. The main text of the manuscript was corrected accordingly.

Minor comments

lines 326-330 ...beyond the scope of this study/work is redundant in the same paragraph

The main text of the manuscript was corrected in line with this comment.

line 511 Pyrodigal should be changed to Prodigal

Pyrodigal was used indeed, see the reference provided.

Accession numbers of assembled MAGs should be included in the main manuscript.

Accession numbers of assembled MAGs are now provided in the main text, chapter *Taxonomic and functional characterisation of the ASC12 consortium*.

Re: mSystems01242-25R2 (Catabolism of Acetosyringone and Co-metabolic Transformation of 2,4,6-Trichlorophenol by a Novel FAD-dependent Monooxygenase)

Dear Prof. Ondrej Uhlík:

Your manuscript has been accepted, and I am forwarding it to the ASM production staff for publication. Your paper will first be checked to make sure all elements meet the technical requirements. ASM staff will contact you if anything needs to be revised before copyediting and production can begin. Otherwise, you will be notified when your proofs are ready to be viewed.

Cover Image Submissions: If you would like to submit a potential Cover Image, please email a file and a short legend to mSystems@asmusa.org. Please note that we can only consider images that (i) the authors created or own and (ii) have not been previously published. By submitting, you agree that the image can be used under the same terms as the published article. Image File requirements: TIF/EPS, 7.5 inches wide by 8.25 inches tall (at least 2,250 pixels wide by 2,475 pixels tall), minimum 300 dpi resolution (600 dpi preferred), RGB, and no figure elements, e.g., arrows or panel labels. The legend should be a short description of the image, 1-2 sentences recommended. Please download and use this interactive template in Adobe to ensure that your proposed cover image meets our size requirements (<https://journals.asm.org/pb-assets/pdf-text-excel-files/ASM-Interactive-Sizing-Cover-Template-1715689791.pdf>).

Sincerely,
Lennart Schada von Borzyskowski
Editor
mSystems